# Machine Learning Analysis of Hyperspectral Images of Damaged Wheat Kernels

**DOI:** 10.3390/s23073523

**Published:** 2023-03-28

**Authors:** Kshitiz Dhakal, Upasana Sivaramakrishnan, Xuemei Zhang, Kassaye Belay, Joseph Oakes, Xing Wei, Song Li

**Affiliations:** 1School of Plant and Environmental Sciences, Virginia Tech, Blacksburg, VA 24061, USA; 2Bradley Department of Electrical and Computer Engineering, Virginia Tech, Blacksburg, VA 24061, USA; 3Graduate Program in Genetics, Bioinformatics and Computational Biology, Virginia Tech, Blacksburg, VA 24061, USA; 4Virginia Tech Eastern Virginia Agricultural Research and Extension Center (AREC), Warsaw, VA 22572, USA; 5Department of Agricultural and Biological Engineering, Purdue University, West Lafayette, IN 47907, USA

**Keywords:** deoxynivalenol (DON), fusarium head blight, hyperspectral imaging, machine learning, object detection

## Abstract

Fusarium head blight (FHB) is a disease of small grains caused by the fungus *Fusarium graminearum*. In this study, we explored the use of hyperspectral imaging (HSI) to evaluate the damage caused by FHB in wheat kernels. We evaluated the use of HSI for disease classification and correlated the damage with the mycotoxin deoxynivalenol (DON) content. Computational analyses were carried out to determine which machine learning methods had the best accuracy to classify different levels of damage in wheat kernel samples. The classes of samples were based on the DON content obtained from Gas Chromatography–Mass Spectrometry (GC-MS). We found that G-Boost, an ensemble method, showed the best performance with 97% accuracy in classifying wheat kernels into different severity levels. Mask R-CNN, an instance segmentation method, was used to segment the wheat kernels from HSI data. The regions of interest (ROIs) obtained from Mask R-CNN achieved a high mAP of 0.97. The results from Mask R-CNN, when combined with the classification method, were able to correlate HSI data with the DON concentration in small grains with an R^2^ of 0.75. Our results show the potential of HSI to quantify DON in wheat kernels in commercial settings such as elevators or mills.

## 1. Introduction

In 2020, wheat was the second most produced cereal crop with a world production of 761 million tons [1]. As a staple food crop [2], the global demand for wheat is increasing due to population growth and changes in dietary regimes [3]. World trade in wheat is greater than all other crops combined [4]. Fusarium head blight (FHB) is one of the most devastating diseases of wheat worldwide [5]. FHB is caused by a fungal plant pathogen, *Fusarium graminearum* [5]. This fungus can infect wheat heads, resulting in significant yield loss [6]. This infection can result in the accumulation of mycotoxins, such as deoxynivalenol (DON), threatening the health of domestic animals and humans [7]. One symptom of FHB is bleaching of the heads shortly after flowering [8]. Kernels infected with the fungus may be shriveled, wrinkled, and lightweight with a dull grayish or pinkish color [9]. These kernels are sometimes referred to as “tombstones” because of their chalky and lifeless appearance [10]. The kernels may be normal in size with slight discoloration yet still harbor mycotoxins even if there is infection at a later stage of kernel development [11]. Kernels contaminated with high levels of DON may be rejected for food and feed [12].

Visual disease-rating methods are based only on the perception of specific colors in visible light spectra. Recent advancements in hyperspectral imaging (HSI) have overcome this limitation of spectral resolution and have seen an increase in their applications in agriculture, including plant phenotyping for disease detection, and assessing chemical content in food and agriculture products [13,14,15]. HSI works by projecting a light beam onto the sample, and the reflectance of the light is collected by an HSI camera. Human vision and typical digital cameras can capture 3 types of color bands (red, 560–580 nm; green, 535–545 nm; and blue, 420–440 nm) in the visible range (VIS spectrum). Commonly used HSI systems can measure changes in reflectance with a spectral range from 350 to 1700+ nm and can see more narrow bands (~7–10 nm bandwidth). A typical hyperspectral scan can generate reflectance data for hundreds of bands. The ability to see extra bands using HSI enables the identification of signatures of chemicals, such as leaf chemical composition [16], because different chemical molecules have different reflectance characteristics. However, this large volume of high-dimensional data has also created challenges in data analysis and in identifying informative wavelengths related to plant health status or chemical composition.

One of the notable usages of hyperspectral sensing is the early identification of anthracnose and gray mold in strawberries, where six machine learning methods were developed, and their classification performance was evaluated and compared [17]. A hyperspectral analysis method based on generative adversarial nets was used for the early detection of the tomato spotted wilt virus, and the results showed that the plant-level classification accuracy was 96.25% before the symptoms were visible [18]. Research on sugar beet for the early detection and differentiation of *Cercospora* leaf spot, leaf rust, and powdery mildew diseases based on support vector machines (SVMs) and spectral vegetation indices showed early differentiation between healthy and inoculated plants, as well as among specific diseases [19]. A study that diagnosed charcoal rot in soybean using hyperspectral imaging and a deep learning model achieved a classification accuracy of 95.73% and an infected class F1 score of 0.87 [20]. However, these methods did not use an HSI camera, which can provide additional spatial resolution of the samples.

A VIS-NIR (400–900 nm) HSI camera can be used to perform a large-scale screen for kernel and flour toxins for small grain producers. Past research has demonstrated that HSI can classify kernels and flour regarding whether DON toxin is above the export limit [21]. The FDA has established advisory levels for DON. The maximum allowable DON level is 1 ppm for finished grain products for human consumption; 5 ppm in swine (not to exceed 20 percent of ration) and all animal species (except cattle and poultry); 10 ppm in ruminating beef and feedlot cattle older than 4 months (providing grain at that level does not exceed 50 percent of diet) and poultry (providing grain at that level does not exceed 50 percent of diet); and 5 ppm in all other animals (providing grain does not exceed 40 percent of diet) [8,22]. Line-scan Raman hyperspectral imaging (RHI) for simultaneous detection of three potential chemical adulterants in wheat flour was carried out by applying spectral angle mapping (SAM) to distinguish the pixels of adulterants from the flour background. The results demonstrated that RHI in combination with SAM performed well in distinguishing the noninvasive quality of powdered foods [23].

Feature extraction and band selection methods are commonly applied to hyperspectral data for dimensional reduction. Some feature extraction methods project the original high-dimensional data into a lower dimension. Band selection methods reduce the dimensionality by selecting a subset of wavelengths from hyperspectral data. A variety of band selection algorithms have been used for plant disease detection, such as the instance-based Relief-F algorithm, genetic algorithms, partial least square, and random forest [24]. In the past several years, applications of machine learning (ML) methods in crop production systems have been increasing rapidly, especially for plant disease detection [25,26,27]. Machine learning refers to computational algorithms that can learn from data and perform classification or clustering tasks, which are suitable for finding the patterns and trends from hyperspectral data. The scikit-learn library provides several functions for different machine learning approaches, dimensional reduction techniques, and feature selection methods [28].

Proper detection and localization for yield components such as determining the distribution of fruits or pods on a plant are important aspects of crop monitoring and for developing robotics and autonomous systems for agriculture applications [29]. Fruit counting and yield estimation are useful not only to farmers but also to breeders and researchers [30,31]. For example, Mask R-CNN (Region-based Convolutional Neural Network) [32] is a convolutional framework for instance segmentation that is simple to train and generalizes well [33]. YOLO [34] is a single-stage network that can detect objects without a previous region proposal stage [35]. In this study, we explored the usage of machine learning methods to analyze HSI data from damaged and healthy kernels. The specific objectives of this study were: (1) To classify symptomatic and non-symptomatic wheat kernels by their spectral reflectance. For this, we tested multiple machine learning methods and then evaluated the classification results. (2) To perform regression analysis to study the correlation between the percentage of symptomatic kernels and pixels with the DON content. (3) To test the Mask R-CNN to segment individual kernels, classify those kernels based on the percentage of symptomatic kernel pixels, count the number of kernels in a sample, and correlate that with the DON content of the sample obtained from GC-MS.

## 2. Materials and Methods

### 2.1. Plant Materials and Experimental Design

A total of 129 wheat cultivars were grown in research plots at the Virginia Tech Eastern Virginia AREC in Warsaw, Virginia during the 2020–2021 growing season. The soil type at this location is Kempsville sandy loam. Meteorological conditions throughout the growing season are presented in Appendix A. These 129 cultivars were planted in a randomized complete block design. Plots (size: 60 × 108 inches) were planted in a conventionally tilled field using a Hege plot planter and were managed using the management practices outlined in Appendix A. Cultivars were grown under normal conditions and inoculated with *F. graminearum*, the causal agent of FHB, at boot stage. Scab epidemics were established by scattering scabby corn grain inoculum during early boot stage (GS45). All plots received overhead fine-mist irrigation for two hours per day for approximately one month after inoculation to stimulate disease development. Plots were harvested using a Wintersteiger Classic combine. Approximately 100 g of grains from each plot was used in this analysis.

### 2.2. DON Content Quantification

Samples were ground to homogeneity with a coffee grinder (Hamilton Beach). DON quantification by Gas Chromatography–Mass Spectrometry (GC-MS) was based on methods described by Tacke [36]. An Agilent 7890B/5977B system was used for GC-MS analysis operating in Selected Ion Monitoring (SIM) mode. The quantification was performed by a small-grain DON testing lab at Virginia Tech. Briefly, the method compares the peak area of DON with a standard and follows several steps such as extraction, cleanup, ion monitoring, and computing DON concentration using the calibration curve. Detailed procedures of DON quantification have been described elsewhere [37].

### 2.3. Image Collection and Data Collection

A total of 200 wheat kernels were placed in a 3D-printed well plate containment with 100 wells (2 kernels per well). Plates were printed to fit a single kernel per well, but due to the size of the plate it was more efficient to have two kernels per well. The dimensions of each well were 7 mm × 3 mm × 3 mm. The well plate was scanned using a benchtop hyperspectral imaging system (Appendix A) Pika L 2.4 (Resonon Inc., Bozeman, MT, USA). All scans were performed using the Spectronon Pro (Resonon Inc., Bozeman, MT, USA) software from a desktop computer connected to the camera system using a USB cable.

Dark current noise was removed before performing the kernel scans using the software. The camera was then calibrated using a white tile (reflectance reference) provided by the manufacturer of the camera system. The white tile was placed in the same conditions where the kernel scans were performed. After each scan, the spectral data of the kernels were collected using post-processing data analysis pipeline in Spectronon Pro. Several areas containing every class of objects were selected using the selection tool, and the mean spectrum was generated. The pixels were chosen manually by randomly selecting from five spectral scans of each class of objects (background, non-symptomatic kernels from the low samples (DON content less than 0.5 ppm), and symptomatic kernels from the severe samples (DON content more than 1.5 ppm)) to avoid any bias. The reflectance data were exported from the software in the form of Excel sheets using the export option. Following the method of estimating the percentage of Fusarium-damaged kernels (FDKs) developed by Ackerman [38], we collected the kernel damage percentage. This method quantifies visual symptoms with the help of human observation.

### 2.4. Data Analysis Pipeline

#### 2.4.1. Classification of Kernels into Symptomatic and Non-Symptomatic Using Machine Learning Methods

Background (plate) and kernel (non-symptomatic and symptomatic) pixels were selected from hyperspectral images, and their reflectance values were imported from Spectronon Pro as input for the analysis pipeline. The reflectance values were normalized, and the average reflectance curve was plotted. Nine different ML methods were deployed on the dataset to compare the accuracy of classification of different data points into different classes of areas. The nine machine learning methods tested are abbreviated as follows: NB = Gaussian Naïve Bayes; KNN = K-nearest neighbors; LDA = linear discriminant analysis; MLPNN = multi-layer perceptron neural network; RF = random forests; SVML = support vector machine with linear kernel; SVMR = support vector machine with radial basis function kernel; G-Boost = gradient boosting; PLSDA = Partial Least Squares Discriminant Analysis. We performed stratified 10-fold cross-validation (CV) by repeating the data points 3 times. Two-class and three-class classifications were performed, and the full analysis pipeline is shown in Figure 1A starting from data collection to segmentation.

For two-class classification, the data points from background and foreground (kernels) were taken, then split into training and testing sets in the ratio of 9:1. The ML methods were trained on the training set, and the predictions were tested on the testing set. The ML methods were evaluated using accuracy, F-1 score, precision, and recall, but the accuracy of the ML methods is reported here for comparison purposes. One of the best ML methods was used to select 200 random kernel pixels from each image that was considered as a low (less than 0.5 ppm), mild (0.5–1.5 ppm), and severe (more than 1.5 ppm) sample as obtained from GC-MS results. These DON contents were quantified from the same sample of 100 g of kernels. There were 10 images of each category such that 6000 pixels (2000 kernel pixels from low images, 2000 mild images, and 2000 kernel pixels from severe images) were used to train 8 machine learning models (except PLSDA among the 9 because of having more than 2 classes) for classification of those pixels into low, medium, and severe classes.

For the three-class classification, the same ML methods were deployed to compare the classification between the three classes (background, non-symptomatic areas, and symptomatic areas for *F. graminearum*). The best ML method was used to select 200 random non-symptomatic kernel pixels from each image that was considered as a low (less than 0.5 ppm) and 200 random symptomatic kernel pixels from each image that was considered as a severe (more than 1.5 ppm) sample. The background pixels were not further used in this case. There were 10 images of each category such that 4000 pixels (2000 from non-symptomatic kernel pixels from low images and 2000 from symptomatic kernel pixels from severe images) were used to train 8 machine learning models for classification of those pixels into non-symptomatic and symptomatic. The evaluation report from one of the best ML methods among the eight methods was taken, and the misclassified and correctly classified pixels were taken for further evaluation. The non-symptomatic pixels were taken as positives, and the symptomatic kernels were considered as negatives to evaluate the data points that were true positives (TPs), true negatives (TNs), false positives (FPs), and false negatives (FNs). These data points were further classified as positives and negatives using the same ML models to evaluate the performance of those eight ML models to classify the data points into TP, TN, FP, and FN. The non-symptomatic pixels predicted as non-symptomatic are referred to as true positives or TPs, the symptomatic pixels predicted as non-symptomatic are referred to as false positives or FPs, the symptomatic pixels predicted as symptomatic are referred to as true negatives or TNs, and the non-symptomatic pixels predicted as symptomatic are referred to as false negatives or FNs.

#### 2.4.2. Regression of Percent Symptomatic Kernels over Total Kernels with GC-MS DON Content

A regression analysis was performed to study the correlation between the percentage of pixels classified as severe and the DON content obtained from GC-MS and the FDK estimates for each image. Finally, a deep neural network method, Mask R-CNN, was implemented to segment individual kernels, classify those kernels based on the percentage of symptomatic kernel pixels, count the number of kernels in a sample, and correlate that with the DON content of the sample obtained from GC-MS. The preprocessing and the analysis of spectral data were performed using the Python programming language (version 3.7.10). Jupyter notebooks for the analysis in this work are provided in the following GitHub repository: https://github.com/LiLabAtVT/WheatHyperSpectral (accessed on 1 March 2023).

For Mask R-CNN, we used a dataset containing 40 RGB images obtained by converting the HSI data into RGB images (each image of size ~800 × 1600). The dataset was split into a training set containing 30 images (60,000 kernels), a validation set composed of 5 images (1000 kernels), and the testing dataset containing 5 images. The evaluation scores were calculated using the testing images. Another dataset consisted of 10 RGB images obtained by cutting the top 2 rows of 10 HSI data and converting them into RGB images (each image of size ~800 × 200). This dataset was split into a training/validation set containing 8 images (28 × 8 kernels) and a test set composed of 2 images (28 × 2 kernels). The objects (kernels) in the images were labeled using VGG Image Annotator (VIA). The annotation files were saved in json format. The different instances of kernels were detected using publicly available Keras/TensorFlow-based implementation for Mask R-CNN by Matterport, Inc. [39] by pre-training with the COCO dataset [40]. Matterport’s implementation of Mask R-CNN for patch-based processing was personalized for our kernel region of interest (ROI) extraction analysis.

A total of 30 hyperspectral (HS) images (10 spectral scans of each class of samples (low (DON content less than 0.5 µg/mL or ppm), mild (DON content between 0.5 and 1.5 µg/mL or ppm), and severe (DON content more than 1.5 µg/mL or ppm))) were further cut into 7 parts to make the kernel detection easier and more precise in the Mask R-CNN analysis. The final dataset contained 210 HS images that were saved as RGB images to be used in the prediction dataset of Mask R-CNN to obtain the ROIs. The extracted ROIs were saved as json file. The same 210 HS images were further used in classifying the pixels into symptomatic and non-symptomatic using the similar protocol described earlier in three-class classification. The ROI information from all the images was imported in the classification results where the kernel pixels were classified into symptomatic and non-symptomatic. The number of kernels that surpassed a threshold percentage of symptomatic kernel pixels were counted as symptomatic kernels. The final count (number of symptomatic kernels) was correlated with DON content obtained from the GC-MS results and the FDK estimates.

## 3. Results and Discussion

### 3.1. Machine Learning of Spectral Reflectance Can Separate Background and Foreground (Kernel)

We first compared nine different machine learning methods for their accuracy in classifying pixels from the background and foreground (kernels). We found a clear separation between the average reflectance of the background and foreground (Figure 2A). The visualization of this result is presented in Figure 1B(II). The average reflectance (normalized intensity) curve for the background remains flat while that of the foreground showed a small peak at 410 nm followed by a small drop and then kept increasing until 600 nm. The average reflectance increases at a slower rate from 600 nm to 700 nm and then increases faster until it starts decreasing from 900 nm onwards.

Most of the ML methods except LDA and NB had ideal performance (100% accuracy) in classifying the kernels into background and foreground, and there were no significant differences between these methods (Figure 2B). LDA and NB were statistically different from each other, and the other remaining 7 ML methods had 100% accuracy with zero variation. The accuracy of NB (0.99 ± 0.02) was very close to the other 7 methods with definite accuracies, and the accuracy of LDA was 89% with 7% variation.

To test whether we can use machine learning methods to classify pixels from kernels with different levels of damage (based on DON content), we performed the following analysis. The kernel samples were separated into three classes: low, mild, and severe samples (see Section 2 for details). To compare the results with the DON content obtained from GC-MS, 200 random kernel pixels were picked from each class of images. Since the accuracy of the classification of SVML in classifying the kernel pixels from the background was ideal, we used SVML to select pixels from kernels first (Figure 2B). Six thousand datapoints were used to calculate the average reflectance curves of the low, mild, and severe classes of images. The average reflectance for each category is different. At a 500 nm to 600 nm wavelength, the low pixels had the highest average reflectance, followed by severe and then mild pixels with the lowest average reflectance among the 3 classes of randomly selected pixels (Figure 2C). The severe category had a lower average reflectance at the 720 to 1050 nm range than the mild category, whereas the mild category had a higher average reflectance below 720 nm. Above 900 nm wavelength, the average reflectance of low pixels remained similar and eventually had a smaller decrement, while that of mild and severe pixels decreased at a faster rate. However, the standard deviation of the curves was very large, and there was no clear separation between them except between 500 nm and 600 nm for the low kernels (Figure 2C).

There were significant differences in accuracies between the eight ML methods of classification to classify the pixels as low, mild, and severe (Figure 2D). G-Boost (0.78 ± 0.02) has the highest accuracy of classification and is significantly different from the other methods. NB (0.59 ± 0.02) has the significantly lowest accuracy of classification. The accuracy of classification of low, mild, and severe pixels using G-Boost was 0.87, 0.69, and 0.77, respectively. This is consistent with the observation that low pixels have more distinct average reflectance (Figure 2C). The confusion matrix of G-Boost results showed that 5349 pixels were correctly classified, and 651 pixels were misclassified. The results suggest that G-Boost can be used to classify the kernel pixels into low, medium, and severe classes with a moderate level of accuracy (0.78).

### 3.2. Three-Class Classification: Background, Non-Symptomatic Area, and Symptomatic Area

In the previous section, we tried to directly classify randomly selected pixels according to the sample DON content (low, mild, and severe). The performance (0.78 accuracy) is not as high as we would expect to use in actual production. By observing the images collected for different samples, we found that not all kernels look the same in the same plate. In severe samples, a lot of kernels appear to be symptomatic, but a fraction of pixels appear to be non-symptomatic (Appendix A). In low DON samples, although most pixels are non-symptomatic, there is a small fraction of pixels that look symptomatic (Appendix A). Because of this observation, we considered whether we could count the foreground pixels with symptoms in each image and use this as a proxy for kernel DON content.

The spectral data collected from the background, non-symptomatic, and symptomatic areas were used to compare the accuracy of classification of different machine learning methods. From the results, we found that the background and kernel curves are well separated (Figure 3A). The average reflectance curve of the background remains similar for all the wavelengths, while that of kernel pixels seems to increase from 350 nm to 900 nm and decreases afterwards. The reflectance curve of the symptomatic area goes up to 400 nm, keeps decreasing until 410 nm, keeps increasing until 900 nm, and then eventually decreases. The reflectance curve of the non-symptomatic area goes up to 400 nm, keeps decreasing to 410 nm, and keeps increasing exponentially to 650 nm. The average reflectance of non-symptomatic areas decreases from 650 nm to 750 nm and then increases faster until it starts decreasing from 900 nm onwards. These results show the applicability of machine learning methods to separate pixels from symptomatic and non-symptomatic areas in the same image. This result is visualized in Figure 1B(III).

To quantify the performance of different models and to select the best performing model, we compared the accuracies of eight different machine learning models. There were significant differences in accuracies in the eight ML methods of classification to classify the background, non-symptomatic area, and symptomatic area (Figure 3B). We found that 5 ML methods had more than 92% accuracy in classifying the kernel HSI into the 3 classes. There were not any significant differences between MLPNN, RF, SVML, and G-Boost, and these methods had significantly higher accuracies than the other methods. The highest accuracy was 0.96 ± 0.03 with SVML, and the lowest was with LDA (0.80 ± 0.09). These results suggest that SVML can be further used to classify the areas of HSI into the background, symptomatic, and non-symptomatic areas.

### 3.3. Correlation of DON Content with Pixel Classification Results

We further picked some random pixels from each image that were considered low (less than 0.5 ppm) and severe (more than 1.5 ppm) and trained machine learning models to classify those pixels into symptomatic and non-symptomatic areas. This ensured that the pixels classified by the models as symptomatic were indeed from samples with high DON content and symptomatic regions. After obtaining the number of pixels of symptomatic and non-symptomatic areas, we developed a matrix that was the percentage of severe pixels and correlated this with the DON content from GC-MS. We did not pick pixels from mild samples (less than 0.5 ppm and more than 1.5 ppm) because we were more interested in the extreme classes.

The average reflectance curves of the low and severe pixels show there is clear separation between them until 410 nm (Figure 3C). The average reflectance curve of the low pixels is higher than that of severe pixels over the entire wavelength range after 410 nm. There were significant differences in accuracies between the eight ML methods of classifying the pixels as low and severe (Figure 3D). G-Boost (0.93 ± 0.01) has the highest accuracy of classification and is significantly different from the other methods, while NB (0.72 ± 0.02) has the significantly lowest accuracy of classification. The accuracy of classification of low and severe pixels using G-Boost was 0.86 and 0.99, respectively. The confusion matrix of the G-Boost results showed that 3851 pixels were correctly classified, and 149 pixels were misclassified.

The misclassified and correctly classified data points were separated from the evaluation report of G-Boost (the best ML method for the data). The non-symptomatic pixels were taken as positives and the symptomatic kernels were considered as negatives to select the data points. The average reflectance curve of those data points was plotted against the wavelengths, and the results showed no clear separation between the TP, TN, FP, and FN pixels (Figure 3E). The average reflectance curve of true positive pixels is higher than that of TN and FP over the entire wavelength range after 410 nm. This result shows that the FP and FN pixels have overlapping reflectance curves and are very noisy. It is difficult to further improve the performance of the ML methods given these reflectance data.

### 3.4. Segmentation of Kernels and Regression of DON Content with Percentage of Severe Kernels

We first carried out the segmentation of kernels using Mask R-CNN in the original RGB images obtained by converting the HSI into RGB images and used Mask R-CNN to count the number of symptomatic kernels. The mean average precision (mAP, IoU = 0.5) for the original RGB images was 0.01, which is low, and the masks were not precise (Appendix A). To improve the performance of this analysis, we cropped the images into smaller images, and the cropped images showed substantially improved precision with mAP of 0.97 (IoU = 0.5). The mean IoU obtained in this case was 0.97, as shown in Appendix A. Using these cropped images for ML and extracting ROIs, we classified a kernel as symptomatic using a different threshold of the percentage of symptomatic pixels over all pixels within a kernel. For each sample, the number of symptomatic kernels was determined, and those numbers were normalized to be used in a correlation analysis with the GC-MS DON content. We correlated the GC-MS DON content with other estimates such as the percentage of severe pixels over all kernel pixels obtained by ML methods and the number of symptomatic kernels obtained by the Mask R-CNN method (Figure 4).

Although a positive correlation was seen between the percentage of severe pixels and the DON content obtained from GC-MS, the coefficient of determination was not more than 0.29 (Figure 4A). To improve this correlation, we first used Mask R-CNN to segment individual kernels, followed by an ML prediction of severely damaged kernel pixels in each kernel, and we then used a threshold to determine whether a kernel was considered “symptomatic” (Table 1). In this table, we show that different thresholds resulted in different R^2^ values. The correlation between the DON content and the number of symptomatic kernels obtained via Mask R-CNN ROIs and the ML method was better than the correlation between the DON content and the percentage of severe pixels, as shown in Figure 4B. The different pixel thresholds for classifying a kernel as symptomatic showed a varying correlation coefficient (Table 1), and the 70% thresholding gave the best result with R^2^ = 0.75.

## 4. Conclusions

In this study, we explored the use of VIS-NIR hyperspectral images to detect the DON content in wheat kernels. G-Boost, an ensemble method, is the most accurate method to classify background and foreground pixels and can more accurately classify the foreground (kernel) pixels into symptomatic and non-symptomatic classes based on the DON contamination level. We demonstrated that deep learning methods combined with classification methods can be used to quantify the DON contamination in small grains. The improved correlations between the GC-MS DON content and the number of symptomatic kernels obtained via Mask R-CNN shows the potential of deep learning methods + HSI to quantify DON in wheat kernels in commercial settings such as using an inline camera on a grain conveyor belt. Additional kernel-wise DON content quantification could help to dissect the connection between DON content and HSI. The kernels were only imaged on one side due to the constraint of the HSI scan. It will be interesting to test the scan on both sides of the kernels and to check whether a double-sided scan can provide additional improvement in the performance of the models.

## Figures and Tables

**Figure 1 sensors-23-03523-f001:**
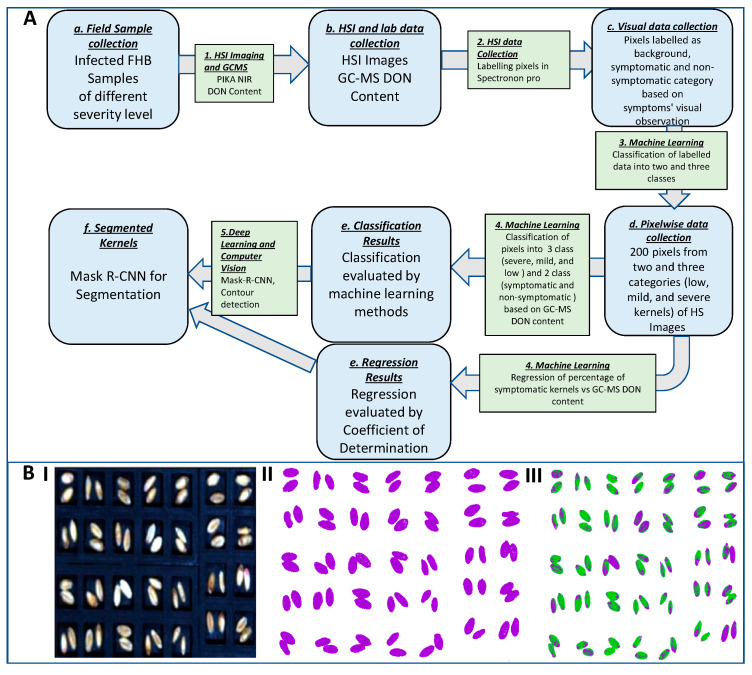
(**A**) Data analysis pipeline for the wavelength selection to classify wheat kernels as symptomatic and non-symptomatic for *Fusarium graminearum.* (**B**) (**I**) RGB representation of HSI. (**II**) Classification of HSI into foreground (purple) and background (white) pixels. (**III**) Classification of HSI into symptomatic (purple), non-symptomatic (green), and background (white) pixels.

**Figure 2 sensors-23-03523-f002:**
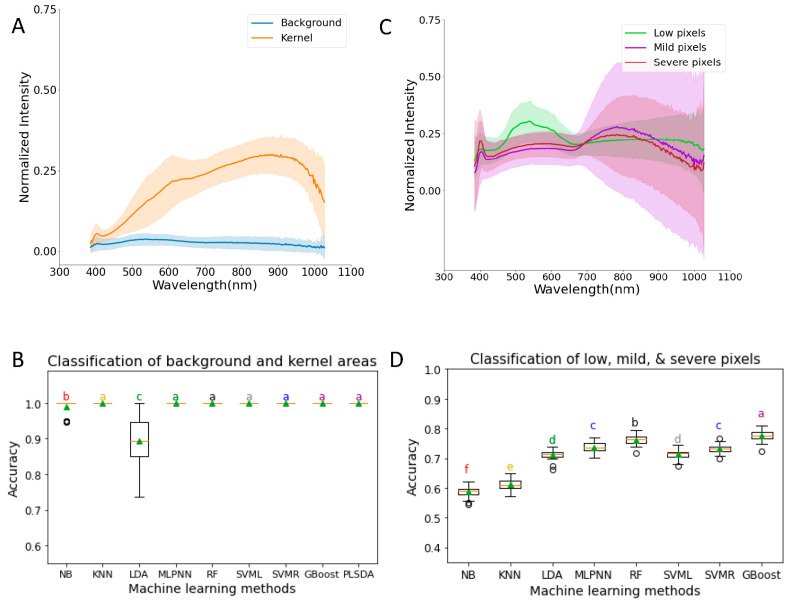
(**A**) Spectral profile of background and kernel areas. (**B**) Comparison of the performance of nine machine learning methods to classify data points into background and kernel areas. (**C**) Spectral profiles of low, mild, and severe pixels. (**D**) Comparison of the performance of eight machine learning methods to classify pixel data points into low, mild, and severe classes. NB = Gaussian Naïve Bayes; KNN = K-nearest neighbors; LDA = linear discriminant analysis; MLPNN = multi-layer perceptron neural network; RF = random forests; SVML = support vector machine with linear kernel; SVMR = support vector machine with radial basis function kernel; G-Boost = gradient boosting; PLSDA = Partial Least Squares Discriminant Analysis. All data in (**A**,**C**) are normalized with a maximum value of 1.0. Letters a–f in (**B**,**D**) indicate statistical test results with same letter representing that the difference is not statistically significant.

**Figure 3 sensors-23-03523-f003:**
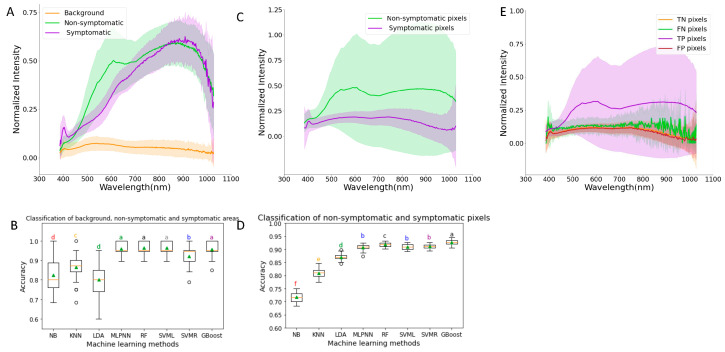
(**A**) Spectral profile of background, non-symptomatic areas, and symptomatic areas of wheat kernel hyperspectral images. (**B**) Comparison of the performance of eight machine learning methods to classify data points into background, non-symptomatic area, and symptomatic area classes. (**C**) Spectral profiles of low and severe pixels. (**D**) Comparison of the performance of eight machine learning methods to classify pixel data points into low and severe classes. (**E**) Spectral profiles of true negative (TN), false negative (FN), true positive (TP), and false positive (FP) pixels. NB = Gaussian Naïve Bayes; KNN = K-nearest neighbors; LDA = linear discriminant analysis; MLPNN = multi-layer perceptron neural network; RF = random forests; SVML = support vector machine with linear kernel; SVMR = support vector machine with radial basis function kernel; G-Boost = gradient boosting. All data in (**A**,**C**,**E**) are normalized with a maximum value of 1.0. Letters a–f in (**B**,**D**) indicate statistical test results with same letter representing that the difference is not statistically significant.

**Figure 4 sensors-23-03523-f004:**
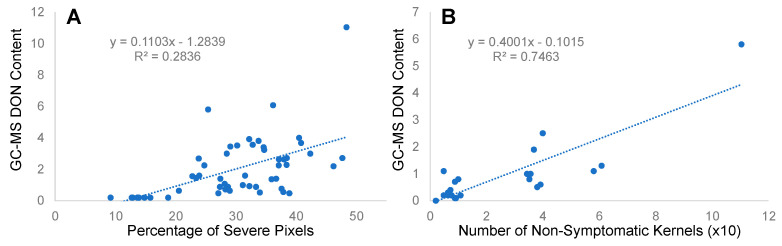
Correlation results between (**A**) percentage of severe pixels and GC-MS DON content and (**B**) GC-MS DON and number of severe kernels (with 70% threshold).

**Table 1 sensors-23-03523-t001:** Coefficient of determination (R^2^) of different thresholding percentages of severe kernel pixels over total kernel pixels to classify a kernel as severe and the GC-MS DON content.

Threshold %	Coefficient of Determination (R^2^)
5	0.23
10	0.31
15	0.36
20	0.44
30	0.57
40	0.62
50	0.73
60	0.74
70	0.75
80	0.74

## Data Availability

The codes and the data are available at Li lab GitHub repository at https://github.com/LiLabAtVT/WheatHyperSpectral (accessed on 1 March 2023).

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
