# Peer review of "Machine Learning Analysis of Hyperspectral Images of Damaged Wheat Kernels"

_sensors, 2023, doi:10.3390/s23073523_

Round 1

Reviewer 1 Report

The manuscript evaluated the ability of HSI for disease classification and correlated the damages with the mycotoxin deoxynivalenol content. The topic of the paper is of interest and the results shown are very interesting from the application point of view. I have some comments that the authors need to address.

1.     The caption of Figure 4 has three subplots. Whereas, there are two subplots in Figure 4.

2.     The DON quantification was by GC-MS. Please describe in detail whether the sample was single kernel ground or multiple kernels ground at the same time, and how to realize the pixel marking of symptomatic areas of the wheat kernel. Some random pixels were picked as low and severe. How to determine the DON quantification corresponding to the pixel point?

3.      “Although positive correlation was seen between the percentage of severe pixel and the DON content obtained from GC-MS, the coefficient of determination was not more than 0.45.” please clarify the reason for the low correlation.

Reviewer 2 Report

The manuscript (sensors-2270023) explored the use of hyperspectral imaging to evaluate damages caused by Fusarium head blight in wheat kernels and correlated the damages with the mycotoxin deoxynivalenol content. G-Boost showed the best performance in classifying wheat kernels into different severity levels, while Mask R-CNN was used to segment wheat kernels from HSI data and achieved a high mAP of 0.97. The study concludes that HSI has the potential to quantify DON in wheat kernels in commercial settings. Introduction, material a methods, results and discussion, conclusion and references (old citations) need major reformulations.

There are references to supplementary tables and figures in manuscript. However, I did not have access to these data. Is this a writing error throughout the manuscript or a file upload error? Please note that some references to figures or supplementary files are far apart from the first citation and their location to be shown in the text.

Please, describe with detail 2.2 section by GC‒MS.

The material and methods section based on training and testing is not adequate, and the writing is not clear. In the image, is it referring to only one biological sample? The description mentions "a test set composed of 5 images (1000 kernels) and the prediction dataset containing 5 images." For clarity, it is important to specify that it is not just one image, but rather one image with multiple samples. Additionally, what is the number of samples analysed? It is not clear if there are 50 different samples in each region of the image or if there are 50 samples in one image. Please clarify this in the text. Just because of this sentence error, the manuscript could be rejected, as the design is not adequate.

The "results and discussion" section were written in a single topic. However, it is not adequate, as the results were not clearly described. In addition, this section has no references. Again, many figures cited in this manuscript do not exist. Figure S7? The English in many sentences is also not adequate. I think that the manuscript needs major changes.

Best regards.

Round 2

Reviewer 1 Report

    • The previous comment has been responded to well. 

    •  

Author Response

Thank you for re-evaluate our submission. 

Reviewer 2 Report

I believe the authors made many modifications compared to the previous version.

However, minor changes are still needed.

-keywords in alphabetical order;

-HSI was defined in the abstract but not in the introduction. Define on line 53;

-Figure 1B in low quality;

-L298. Why this error of 0.02? Rounding, because this represents 101%!?

-Figure 3. What is the peak of normalization in each figure? The same, correct?

Best regards
